# Clinical Outcome of Colorectal Cancer Patients with Concomitant Hypertension: A Systematic Review and Meta-Analysis

**DOI:** 10.3390/jpm14050520

**Published:** 2024-05-14

**Authors:** Daniel Sur, Constantin Ionut Coroama, Alessandro Audisio, Roberta Fazio, Maria Coroama, Cristian Virgil Lungulescu

**Affiliations:** 1Department of Medical Oncology, University of Medicine and Pharmacy “Iuliu Haţieganu”, 400012 Cluj-Napoca, Romania; daniel.sur@umfcluj.ro; 2Department of Medical Oncology, The Oncology Institute “Prof. Dr. Ion Chiricuţă”, 400015 Cluj-Napoca, Romania; 3Department of Digestive Oncology, Institute Jules Bordet, The Brussels University Hospital, 1070 Brussels, Belgium; alessandro.audisio@hubruxelles.be (A.A.); roberta.fazio@cancercenter.humanitas.it (R.F.); 4Department of Pathophysiology, Faculty of Medicine, “Iuliu Hațieganu” University of Medicine and Pharmacy, 4–6 Victor Babeş Street, 400012 Cluj-Napoca, Romania; ilea_maria@elearn.umfcluj.ro; 5Department of Cardiology, “Niculae Stancioiu” Heart Institute, 400001 Cluj-Napoca, Romania; 6Department of Medical Oncology, University of Medicine and Pharmacy Craiova, 200349 Craiova, Romania; cristian.lungulescu@umfcv.ro

**Keywords:** colorectal cancer, hypertension, survival, systematic review, meta-analysis

## Abstract

Background: Arterial hypertension is regarded as a possible biomarker of treatment efficacy in colorectal cancer. Also, extended anti-angiogenic use in the metastatic treatment of the colorectal neoplasm may result in elevated blood pressure. We carried out a systematic review and meta-analysis to assess the clinical outcome of colorectal cancer patients with concomitant hypertension (HTN). Methods: We conducted a systematic search on Embase, Web of Science, Scopus, PubMed (Medline), the Cochrane Library, and CINAHL from inception until October 2023 for articles that addressed the relationship between HTN and progressive free survival (PFS), overall survival (OS), and overall response rate (ORR) for the first and second line of systemic therapy in patients with metastatic colorectal cancer. Results: Eligibility criteria were met by 16 articles out of 802 screened studies. Pooled analysis showed that HTN was associated with significantly improved PFS (HR: 0.507, 95% CI: 0.460–0.558, *p* ≤ 0.001) and OS (HR: 0.677, 95% CI: 0.592–0.774, *p* ≤ 0.001) in patients with metastatic colorectal cancer. In addition, the pooled RR of HTN for the ORR (RR: 1.28, 95% CI: 1.108–1.495, *p* = 0.001) suggests that HTN could be a predictive factor of ORR in patients with metastatic colorectal cancer. Conclusions: Elevated blood pressure is associated with better clinical outcomes in patients with metastatic colorectal cancer.

## 1. Introduction

Colorectal cancer is a common neoplasia, currently the second most incident malignancy. According to the Global Burden of Diseases, Injuries, and Risk Factors Study (GBD), 2.17 million cases and 1.09 million deaths globally were attributed to colorectal cancer in 2019 [1]. It is known that the identification of biomarkers in order to predict the clinical outcome represents a real challenge. Hypertension (HTN) may have an important prognostic value as it shares overlapping risk factors and pathophysiological mechanisms with cancer, such as inflammation, oxidative stress, cigarette smoking, alcohol consumption, unhealthy diet, and physical inactivity [2].

The introduction of anti-angiogenic agents into the therapeutic regimens for cancer reignited the interest in arterial HTN as a prognostic indicator in the clinical outcome of colorectal cancer patients [3]. The inhibitors of the vascular endothelial growth factor (VEGF) signalling pathway (bevacizumab, sorafenib, sunitinib, and pazopanib) act via multiple mechanisms similar to the pathophysiology of preeclampsia, influencing nitric oxide production in the arterial wall, and their effect on blood pressure is considered as a surrogate for anti-cancer treatment efficacy [4]. An increased blood pressure has been reported in up to 30% of patients under treatment with VEGF inhibitors. Treatment-related HTN usually occurs in the first month of initiating the anti-cancer therapy and stabilises after completing the first treatment cycle [5].

Therefore, optimising anti-hypertensive therapy should be considered in patients developing HTN (>140/90 mmHg) or presenting an increased diastolic blood pressure of more than 20 mmHg compared with the pre-treatment values. In this context, renin-angiotensin-system (RAS) blockers and CCBs are the preferred drugs, and a pharmacological therapeutic combination strategy is frequently needed [6]. Although there is no consensus on withholding oncological treatment in case of an increase in blood pressure, the 2023 European Society for HTN (ESH) guidelines recommend its temporary discontinuation in patients who are symptomatic or present a Grade 3 HTN [2]. In such cases, controlling the blood pressure and symptoms represents a priority so that anti-cancer treatment can be initiated as soon as possible [2].

Nevertheless, the bidirectional cause–effect relationship between HTN and cancer is still a matter of debate. For instance, although arterial HTN is the most common cardiovascular comorbidity reported in cancer registries, with elevated blood pressure reported in more than one-third of patients [7], some cancer therapies may cause resistant HTN due to their pressor effect [2]. In addition, elevated diastolic blood pressure has been proposed as an independent risk factor for renal cell carcinoma, but the relationship between HTN and other cancers is still unclear [8].

Moreover, an earlier meta-analysis demonstrated no significant association between angiotensin-converting enzyme inhibitors (ACEIs), angiotensin II receptor antagonists (ARBs), beta-blockers (BBs), thiazides, and cancer risk [9]. However, the evidence was insufficient to completely rule out the increased risk, particularly with calcium channel blockers (CCBs), considering that CCBs were associated with an increased risk of prostate and skin cancer with a small effect size for all other types of cancers [9]. It should be noted that analysing data related to elevated blood pressure in cancer trials can be challenging due to the variability in the definitions of HTN, which can be classified according to various versions of the common terminology criteria for adverse events (CTC AE) in cancer therapy [2].

Therefore, to assess the prognostic value of HTN in patients with colorectal cancer, this systematic review and meta-analysis aimed to evaluate the clinical outcome of patients with concomitant HTN.

## 2. Materials and Methods

The protocol of this systematic review and meta-analysis study was prospectively registered at PROSPERO [CRD42022320971]. The study was carried out according to the preferred reporting items for systematic reviews and meta-analyses (PRISMA) guidelines [10] and written following the meta-analysis of observational studies in epidemiology (MOOSE) proposal [11].

### 2.1. Data Sources and Search Strategy

A systematic search was conducted on Embase, Web of Science, Scopus, PubMed (Medline), the Cochrane Library, and CINAHL from inception until October 2023 by two independent authors using a combination of keywords and medical subject headings (MeSH) (Appendix A) as follows: (((((((((“HTN”[Mesh]) OR “Blood Pressure”[Mesh]) OR “Diastole”[Mesh]) OR “Systole”[Mesh]) AND “Colorectal Neoplasms”[Mesh]) OR “Rectal Neoplasms”[Mesh]) AND “Survival”[Mesh]) OR “Mortality”[Mesh]) OR “Disease-Free Survival”[Mesh]). The cross-references from the selected studies were further searched for additional articles. Articles identified through forward/backward search were screened and evaluated using the same study selection criteria.

### 2.2. Selection Criteria

Relevant articles were screened by title and abstract after removing duplicates. Studies were eligible for inclusion if they addressed the relationship between HTN and progressive free survival, overall survival, and overall response rate for the first and second line of systemic therapy in patients with metastatic colorectal cancer. The selected studies were then examined in full text to confirm eligibility.

The inclusion criteria for articles were as follows: (1) observational studies (non-randomised studies), including retrospective and prospective studies, reporting the relationship between HTN (Grade > 1, according to the CTC AE) and progressive free survival, overall survival, and overall response rate in patients with metastatic colorectal cancer under treatment with chemotherapy +/− targeted therapies; (2) median follow-up of at least one year; (3) publications reporting sufficient information on the overall response rate or its associated odds ratio, or hazard ratio for progression-free survival or overall survival; and (4) studies published as original articles. The exclusion criteria were as follows: (1) full text available electronically; (2) publication in a language other than English; (3) comments, letters, editorials, protocols, guidelines, and review papers; and (4) studies with insufficient outcome data.

Two independent authors assessed the eligibility of all potential articles according to the above criteria. In the case of disagreements, a third author was consulted.

### 2.3. Data Extraction

Two independent authors retrieved information from the eligible articles following the inclusion and exclusion criteria. Data was collected on a standardised data sheet that included the following: (1) study ID (name of first author, year of publication), (2) country of study, (3) study design, (4) number, age and gender of participants, (5) primary tumour location, (6) number of cases of liver metastases vs. no liver metastases, (7) number of patients with HTN, (8) HTN criteria used, (9) HTN cut-off point in controls, and (9) outcomes measures. A third author checked the datasheet for accuracy.

### 2.4. Quality Assessment

The Newcastle–Ottawa Scale (NOS) was used to assess the quality of non-randomised studies, which evaluates selection bias, the comparability of the exposed and control participants, and outcome evaluation [12]. The NOS comprises three sections with a maximum score of 9 points: (1) selection of exposed (patients with HTN) and control groups (maximum 4 points), (2) comparability of study groups (maximum 2 points), and (3) evaluation of outcomes (maximum 3 points). Two independent authors assessed quality independently, and discordances were solved by discussion. The quality of each study was rated using the following scoring algorithms: ≥7 points were considered “good”, 2 to 6 points were considered “fair”, and ≤1 point was considered “poor” quality [12].

### 2.5. Outcome Measures

The primary outcome was progression-free survival (PFS), which represents the time from randomisation until disease progression or death from any cause, whichever occurs first. Secondary outcomes were overall survival (OS), defined as the time between the start of randomisation to date of death due to any cause and overall response rate (ORR), which is the sum of partial and complete response rates according to the Response Evaluation Criteria in Solid Tumours v1.1, which is a guideline that describes a standard approach to solid tumour measurement and definitions for the objective assessment of change in tumour size for use in adult and paediatric cancer clinical trials [13].

HTN was defined according to the CTC AE of the National Cancer Institute [14]. Grade 1 represents an asymptomatic transient (24 h) increase of more than 20 mmHg (diastolic) or greater than 150/100 mmHg if previously within the normal range and no intervention was indicated. Grade 2 represents a recurrent, persistent (24 h), or symptomatic increase of more than 20 mmHg (diastolic) or greater than 150/100 mmHg if previously within the normal range and managed by monotherapy. Grade 3 classification requires HTN management using more than one drug or more intensive therapy than Grade 2. Finally, Grade 4 represents a hypertensive crisis.

### 2.6. Statistical Analysis

The statistical analyses were performed using Comprehensive Meta-Analysis version 3 (Biostat Inc., Englewood, NJ, USA). The risk ratios (RRs) with 95% confidence intervals (Cis) for ORR, as well as the hazard ratio (HR) with 95% CI for PFS or OS, were calculated for patients with metastatic colorectal cancer with and without HTN (Grade > 1, according to CTC AE). A *p*-value of <0.05 was considered as the threshold for statistical significance.

The Cochrane chi-squared test was used to evaluate heterogeneity among articles, with *p*-value < 0.05 indicating the existence of heterogeneity. The I^2^ value was calculated to estimate heterogeneity’s impact on the meta-analysis. I^2^ values ≥ 50% and *p* < 0.05 indicated a moderate to high heterogeneity among pooled studies. A random-effects model was adopted. We also performed subgroup and sensitivity analysis to assess the possible source of heterogeneity.

Publication bias was assessed by visually examining the symmetry in funnel plots. In addition, Egger’s test (weighted regression test) was used to assess publication bias statistically [15]. The Duval and Tweedie nonparametric trim-and-fill methods were performed to further assess the potential publication bias [16] using Statistical Package for Social Sciences (SPSS) version 25 (SPSS Inc., Chicago, IL, USA).

## 3. Results

### 3.1. Identification of Studies

The database search yielded 802 studies for screening (PubMed (*n* = 425), Embase (*n* = 124), Web sciences (*n* = 107), Scopus (*n* = 87), Cochrane Library (*n* = 34), CINAHL (*n* = 25)), of which 298 abstracts were identified as potentially eligible and retrieved for full-text review. Finally, 16 articles met the eligibility criteria and were included in this systematic review and meta-analysis [3,17,18,19,20,21,22,23,24,25,26,27,28,29,30,31]. The detailed PRISMA flowchart for study screening and selection is presented in Figure 1.

### 3.2. Characteristics of Included Studies

All the included articles were published between 2009 and 2023 and originated from eight countries. Among the 16 articles included in this systematic review and meta-analysis, 15 were retrospective [3,17,18,19,20,21,22,24,25,26,27,28,29,30,31], and only 1 was a prospective study [23]. The sample size of the included articles varied from 45 to 750 participants, with a slightly higher proportion of male patients (58%). In total, 66% of the primary tumours were in the colon, while 34% were in the rectum. HTN was diagnosed using the CTC AE version 2.0, 3.0, 4.0, or 5.0 in 15/16 studies. The characteristics of the included studies are summarised in Table 1. 

### 3.3. Quality Assessment

The included studies had a median NOS score of eight, with a maximum score of nine and a minimum of six. In total, twelve articles were evaluated to be of good quality (score ≥ 7) [3,18,19,21,22,23,24,25,27,29,30,31], while four articles were assessed to be of fair quality (score = 6) [17,20,26,28]. Table 2 summarises the quality assessment scores for the observational studies.

Eight of the sixteen included studies had high scores in the selection section. On the other hand, all articles had a high representativeness of their samples. Regarding comparability, all included studies described a statistical analysis to compare the HTN and non-HTN groups. However, eleven studies controlled both groups for the outcomes and additional factors (e.g., age) and scored two stars. Finally, regarding outcomes, all included studies adequately described the assessment of the outcome and scored one star. All studies scored a supplementary star as they were followed up after an adequate amount of time, while the follow-up cohort rate was adequate in 10/16 of the studies.

### 3.4. Progression-Free Survival

Thirteen studies reported the PFS outcome with high heterogeneity (Chi^2^ = 53.29, *p* ≤ 0.001, I2 = 77%), so a random effect model was used. The forest plot analysis showed that the pooled HR of HTN was significantly below 1 (HR: 0.586, 95% CI: 0.468–0.733, *p* = 0.000), suggesting that HTN was associated with improved PFS in patients with metastatic colorectal cancer (Figure 2).

### 3.5. Overall Survival

Twelve studies reported the OS outcome, albeit with heterogeneity (Chi^2^ = 26.16, *p* = 0.006, I^2^ = 57%). An analysis of the pooled data showed that HTN was associated with significant improvement in the OS (HR: 0.633, 95% CI: 0.504–0.795, *p* = 0.000) of patients with metastatic colorectal cancer (Figure 3).

### 3.6. Overall Response Rate

Nine studies reported the ORR outcome with heterogeneity (Chi^2^ = 25.44, *p* = 0.001, I^2^ = 68%). The pooled risk ratio of HTN for the ORR was significantly above 1 (RR: 1.439, 95% CI: 1.085–1.909, *p* = 0.011), suggesting that HTN could be a predictive factor of ORR in patients with metastatic colorectal cancer (Figure 4).

### 3.7. Publication Bias

Asymmetry was observed in the visual inspection of the funnel plots for PFS (Figure 5a) and ORR (Figure 5b), indicating publication bias. Egger’s regression test confirmed publication bias for PFS (*p* = 0.02) and ORR (*p* = 0.03) outcomes. However, we used the trim-and-fill method to correct the bias, which did not alter the significant association between HTN and both PFS and ORR.

On the other side, no publication bias was detected for the OS outcome (*p* = 0.24), which showed a symmetric funnel (Figure 5c).

### 3.8. Subgroup Analysis

For the PFS outcome, subgroup analysis showed that the geographic origin of studies, sample size, HTN criteria, and HTN grade of controls were significant sources of heterogeneity based on the test for subgroup heterogeneity (*p* < 0.05) (Table 3).

For the OS outcome, the geographic origin of studies and sample size were identified as significant confounders for the prediction of HTN and a possible cause of heterogeneity, given the differences in HR between subgroups (*p* < 0.05). When the analysis was restricted to studies conducted in the Americas, the HR of overall survival for HTN was statistically higher (HR: 0.861, 95%CI: 0.714–1.039) than in studies conducted in Europe (HR: 0.582, 95%CI: 0.476–0.713) or Asia (HR: 0.349, 95%CI: 0.212–0.575). However, the HTN criteria and HTN cut-off point in controls were not sources of heterogeneity (*p* > 0.05) (Table 3).

In subgroup analysis for ORR, only HTN criteria may account for significant heterogeneity based on significant differences in the stratified RR (*p* = 0.001). When the analysis was restricted to studies with CTC AE V4.0 (RR: 3.970, 95%CI: 1.981–7.955), the RR of the overall response rate of CRC patients for HTN was over twice and thrice as much as that in studies with CTC AE V2.0 (RR: 1.649, 95%CI 1.110–2.452) and CTC AE V3.0 (RR: 1.155, 95%CI: 0.978–1.364), respectively (Table 3).

### 3.9. Sensitivity Analysis

A leave-one-out sensitivity analysis was performed to identify further the possible source of heterogeneity in the pooled HR and RR of HTN for the PFS, OS, and ORR outcomes in patients with metastatic colorectal cancer. A stable trend in the risk estimates was noted for the three outcomes, indicating the strong reliability of the meta-analysis. The HR of HTN for the PFS and OS outcomes ranged from 0.467 [95% CI: 0.421, 0.518] to 0.630 [95% CI: 0.560, 0.708] and from 0.609 [95% CI: 0.522, 0.710] to 0.709 [95% CI: 0.616, 0.816], respectively. Similarly, the RR of HTN for the ORR outcome ranged from 1.219 [95% CI: 1.040, 1.428] to 1.434 [95% CI: 1.210, 1.700] (Table 4).

## 4. Discussion

The current systematic review and meta-analysis, including 16 studies, found that HTN is associated with improved PFS and OS in patients with metastatic colorectal cancer. Moreover, HTN could be a predictive factor for ORR.

Our results are consistent with a previous systematic review by Lombardi et al. [32] which demonstrated a positive prognostic value of bevacizumab-related HTN on PFS and OS outcomes in colorectal cancer patients. However, there was no difference in PFS or OS in the subgroup analysis between patients with baseline HTN and patients who never experienced more than Grade 2 HTN based on CTC AE version 3 [32].

Furthermore, Xuan et al. [33] conducted a meta-analysis of 25 studies and demonstrated that hypertensive male patients had an increased risk of colorectal cancer. Although our analysis does not have any overlapping objective with the study by Xuan et al. [33], it is important to emphasise that HTN is positively correlated with CRC risk and can bring more information about the development of CRC in hypertensive patients.

There are at least two possible explanations for the association between HTN and colorectal cancer in terms of PFS or OS. Firstly, it has been repeatedly shown that HTN may represent a clinical biomarker for the efficacy of anti-angiogenic agents in colorectal cancer treatment [3,17,18,19,21,22,25,28,29,31]. It has been postulated that anti-angiogenic treatment induces increased vascular resistance and subsequent arterial HTN by decreasing the production of endothelial cell-derived nitric oxide [34]. Secondly, the improved survival in hypertensive colorectal cancer patients could also be hypothesized to result from the efficacy of anti-hypertensive treatment. Unfortunately, comprehensive details of the anti-hypertensive treatment regimen used in these studies, such as posology or treatment duration, are scarcely reported. However, it is widely known that the common anti-hypertensive drugs decrease cardiovascular risk and all-cause mortality along with blood pressure [6]. For instance, two meta-analytic studies have demonstrated that a 10 mmHg reduction in systolic blood pressure or a 5 mmHg reduction in diastolic blood pressure is associated with significant reductions in all major cardiovascular events by 20% and all-cause mortality by 10–15% [35,36].

This study has several limitations that deserve further discussion. First, high heterogeneity was detected between the analysed studies regarding the year of publication, population demographics, different versions of HTN classification, and measurement and adjustment for confounders. These differences could not be completely eliminated despite using the appropriate meta-analytic techniques. Also, it is worth noting that 15 of the 16 included studies had a retrospective design, thus constituting an additional limitation of this meta-analysis. Moreover, the articles were published between 2009 and 2023, and thirteen out of sixteen studies were carried out before 2020. During these years, new developments have been made in treating colorectal cancer and arterial HTN.

Given that most publications focused on the prognostic role of HTN related to cancer treatment, more prospective studies are needed to assess the impact of pre-existing HTN on clinical outcomes in colorectal cancer. A noteworthy consideration is that a substantial number of studies analysed only the impact of bevacizumab-related HTN, whilst other anti-angiogenic agents are less investigated. The novelty of our study approach consists in not limiting the evaluation to treatment-related HTN. Furthermore, patients with active cancer were usually excluded from randomised controlled clinical trials in HTN. Conversely, patients with uncontrolled HTN or elevated blood pressure were not included in cancer trials. Thereby, there is no clear evidence available to guide the most appropriate management and drug therapy for HTN in patients with cancer.

## 5. Conclusions

In conclusion, this systematic review and meta-analysis showed that elevated blood pressure is associated with better clinical outcomes in patients with metastatic colorectal cancer. Our results are consistent with other meta-analyses on this subject, but more prospective studies are needed to investigate the prognostic role of non-oncological treatment-related HTN. In addition to the positive impact on progression-free survival and overall survival, our meta-analysis showed that arterial HTN may be a predictive clinical biomarker for the overall response rate in colorectal cancer patients.

## Figures and Tables

**Figure 1 jpm-14-00520-f001:**
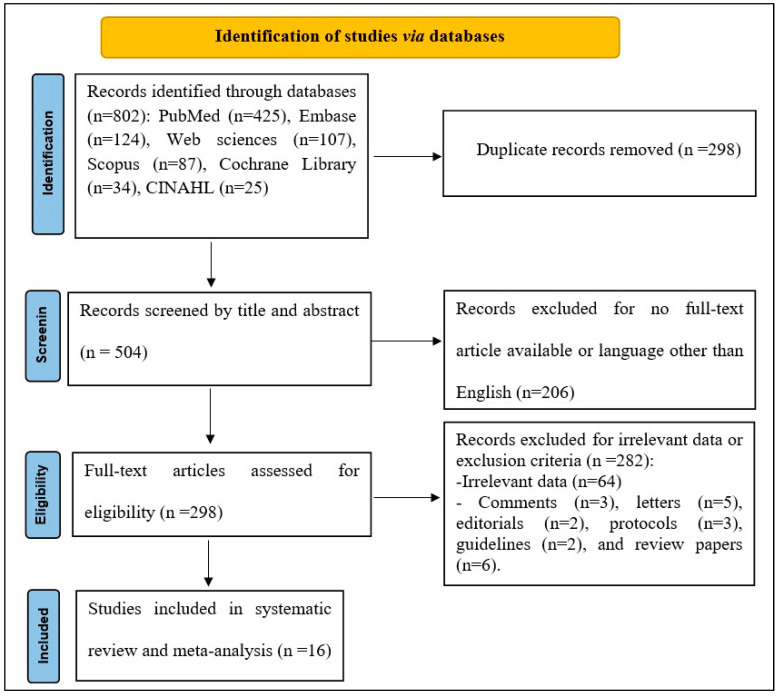
PRISMA flowchart of the study analysis.

**Figure 2 jpm-14-00520-f002:**
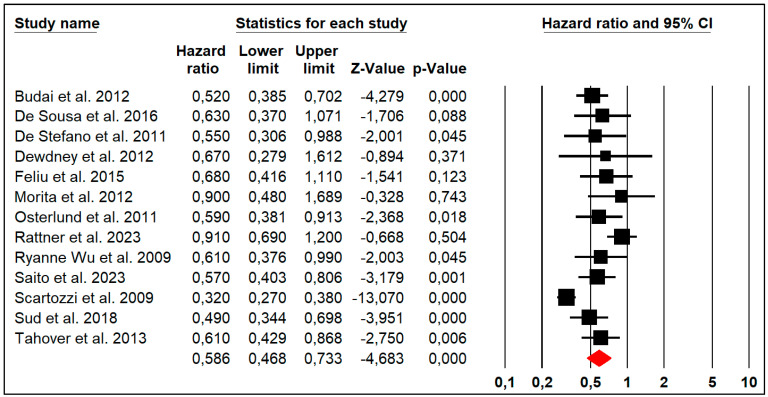
Forest plot of the pooled hazard ratio of HTN for progression-free survival in patients with metastatic colorectal cancer.

**Figure 3 jpm-14-00520-f003:**
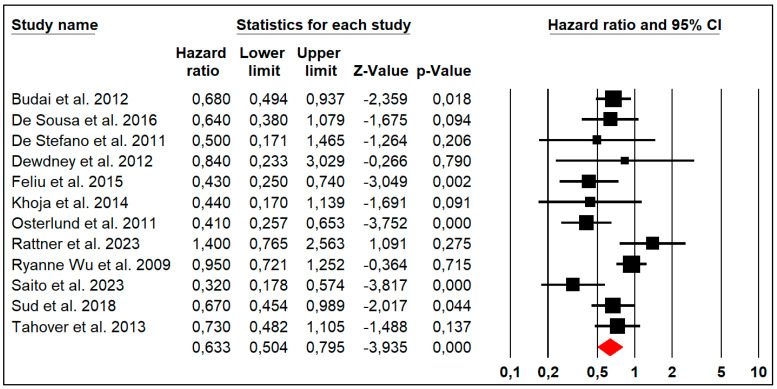
Forest plots of the pooled hazard ratio of HTN for overall survival in patients with metastatic colorectal cancer.

**Figure 4 jpm-14-00520-f004:**
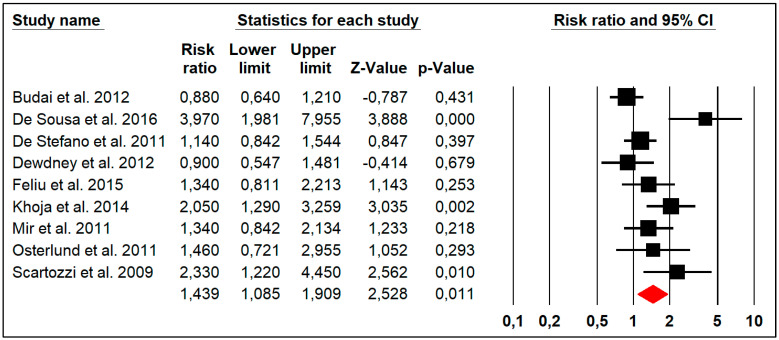
Forest plots of the pooled risk ratio of HTN for the overall response rate in patients with metastatic colorectal cancer.

**Figure 5 jpm-14-00520-f005:**
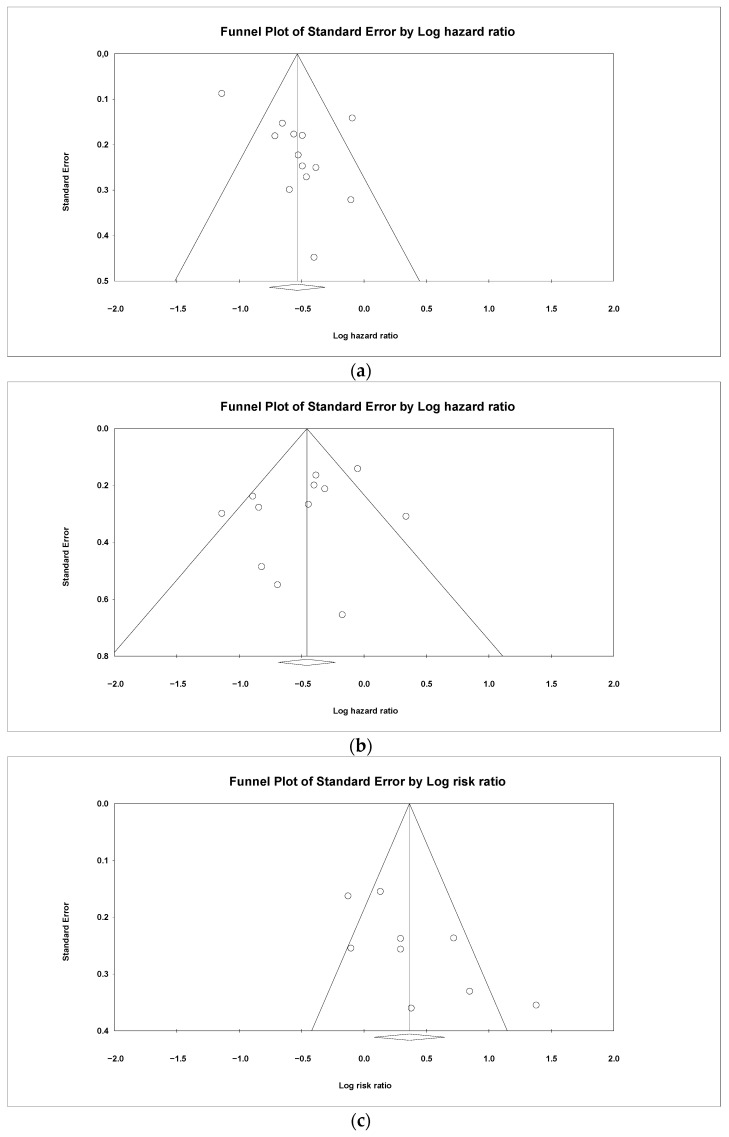
Funnel plots of (**a**) progression-free survival, (**b**) overall survival, and (**c**) overall response rate outcomes for HTN in patients with metastatic colorectal cancer.

**Table 1 jpm-14-00520-t001:** Characteristics of the included studies.

Author (Year) Country	Sample Size	Mean Age	Gender (M/F)	Primary Tumour Location	Liver Metastases vs. No Metastases	Patients with HTN *n* (%)	Oncological Treatment	HTN Criteria	HTN Cut-Off in Controls	Median PFS (Months) HTN/Non-HTN	Median OS (Months) HTN/Non-HTN	ORR (%) HTN/Non-HTN
Budai et al. (2013) [17] Hungary	232	56.5	126/106	Colon: 90 Rectum: 142	315/135	NA	Bevacizumab + mFOLFIRI	CTC AE V3.0	Grade ≤ 1	NA	NA	NA
Dionisio de Sousa et al. (2016) [18] Portugal	79	60.3	53/26	Colon: 50 Rectum: 29	58/43	41 (51.9)	Bevacizumab + FOLFOX or FOLFIRI (1st line treatment)	CTC AE V4.0	Grade ≤ 1	NA	33/21	NA
De Stefano et al. (2011) [19] Italy	74	58	42/32	Colon: ND Rectum: ND	38/36	13 (17.6)	Bevacizumab + FOLFIRI or XELIRI or FOLFOX or XELOX or FOLFOXIRI	CTC AE V3.0	Grade = 0	15.1/8.3	35.5/26.7	84.6/42.6
Dewdney et al. (2012) [20] UK	45	NA	NA	Colon: ND Rectum: ND	ND	7 (15.6)	Bevacizumab + XELOX (before and after liver metastasis resection)	CTC AE V3.0	Grade = 0	NA	NA	71/78
Feliu et al. (2015) [21] Italy	127	76	78/49	Colon: ND Rectum: ND	ND	20 (15.7)	Bevacizumab + capecitabine or XELOX	CTC AE V2.0	Grade = 0	NA	NA/16.9	NA
Khoja et al. (2014) [22] UK	50	61	NA	Colon: ND Rectum: ND	40/51	7 (14)	Bevacizumab or TKI + chemotherapy	CTC AE V3.0	Grade ≤ 1	10.9/9.4	25.2/21.6	NA
Mir et al. (2011) [23] France *	119	61	63/56	Colon: ND Rectum: ND	ND	65 (54.6)	Bevacizumab + 5-FU combination chemotherapy	CTC AE V3.0	Grade = 0	NA	NA	76.9/79.6
Morita et al. (2013) [24] Japan	60	62	38/22	Colon: ND Rectum: ND	ND	16 (26.7)	Bevacizumab + chemotherapy (mFOLFOX6 or FOLFIRI or LV5FU2 or XELOX)	CTC AE V4.0	Grade ≤ 2	NA	NA	NA
Osterlund et al. (2011) [25] UK	101	59	54/47	Colon: 56 Rectum: 45	ND	57 (56.4)	Bevacizumab + chemotherapy (FOLFIRI or XELIRI or irinotecan or oxaliplatin-based therapy or 5-FU-based therapy)	CTC AE V3.0	Grade = 0	10.5/5.3	25.8/11.7	52.6/45.5
Ottaiano et al. (2023) [26] Italy	244	64	127/117	Colon: ND Rectum: ND	ND	110 (45.1)	Bevacizumab + chemotherapy	ACC/AHA	Normal, less than 120/80 mm Hg	NA	26/42	NA
Rattner et al. (2023) [27] Canada	750	63.3	481/268	Colon: 438 Rectum: 165 Colon and rectum: 146	572/886	127 (17)	Cetuximab ± brivantinib (TKI)	CTC AE V3.0	Grade = 0	3.65/3.71	8.9/7.8	NA
Ryanne Wu et al. (2009) [28] USA	84	NA	42/42	Colon: NA Rectum: NA	NA	36 (42.9)	Bevacizumab + chemotherapy	CTC AE V3.0	Grade = 0	NA	NA	NA
Saito et al. (2023) [29] Japan	100	65	56/44	Colon: 56 Rectum: 35	66/100	30 (30)	Regorafenib	CTC AE V5.0	Grade < 2	53/56 days	205/187 days	NA
Scartozzi et al. (2009) [3] Italy	84	55.5	25/14	Colon: 30 Rectum: 9	32/25	8 (20.5)	Bevacizumab + FOLFIRI (only 1st line treatment)	CTC AE V2.0	Grade < 2	14.5/3.1	NA/15.1	75/32
Sud et al. (2018) [30] Canada	572	64.2	368/204	Colon: 332 Rectum: 133 Colon and rectum: 107	ND	149 (26)	Cetuximab	NA	NA	3.5/1.8	7.3/5.7	NA
Tahover et al. (2013) [31] USA	181	61.82	95/86	Colon: 125 Rectum: 56	ND	81 (44.8)	Bevacizumab + 1st or 2nd line chemotherapy	CTC AE V4.0	Grade ≤ 1	17.2/29.9	36.8/NA	NA

* All included except Mir et al. (2011) [23] were retrospective studies. (m)FOLFIRI: modified—leucovorin calcium (folinic acid), fluorouracil, and irinotecan, (m)FOLFOX: modified—leucovorin calcium (folinic acid), fluorouracil, and oxaliplatin, ACC/AHA: American College of Cardiology/American Heart Association, CTC AE: common terminology criteria for adverse events, FOLFOXIRI: folinic acid, 5-fluorouracil, oxaliplatin, and irinotecan, LV5FU2: leucovorin calcium (folinic acid) and fluorouracil, ND: not defined, TKI: tyrosine kinase inhibitors, XELIRI: capecitabine and irinotecan, XELOX or CAPOX: oxaliplatin and capecitabine.

**Table 2 jpm-14-00520-t002:** Newcastle–Ottawa quality assessment scale for observational studies included in the meta-analysis.

Study	Selection	Comparability	Outcome	Total Quality Score	Quality Assessment
Budai et al. (2013) [17]	★★★★	★	★★	6	Fair
Dionisio de Sousa et al. (2016) [18]	★★★★	★★	★★★	9	Good
De Stefano et al. (2011) [19]	★★★	★★	★★★	8	Good
Dewdney et al. (2012) [20]	★★★	★	★★	6	Fair
Feliu et al. (2015) [21]	★★★★	★★	★★	8	Good
Khoja et al. (2014) [22]	★★★★	★★	★★	8	Good
Mir et al. (2011) [23]	★★★	★★	★★★	8	Good
Morita et al. (2013) [24]	★★★	★★	★★	7	Good
Osterlund et al. (2011) [25]	★★★★	★★	★★★	9	Good
Ottaiano et al. (2023) [26]	★★	★	★★★	6	Fair
Rattner et al. (2023) [27]	★★★★	★★	★★★	9	Good
Ryanne Wu et al. (2009) [28]	★★★	★	★★	6	Fair
Saito et al. (2023) [29]	★★★★	★★	★★★	9	Good
Scartozzi et al. (2009) [3]	★★★	★★	★★★	8	Good
Sud et al. (2018) [30]	★★★	★	★★★	7	Good
Tahover et al. (2013) [31]	★★★★	★★	★★★	9	Good

**Table 3 jpm-14-00520-t003:** Subgroup analyses of progressive free survival, overall survival, and overall response rate outcomes for the presence of hypertension in patients with metastatic colorectal cancer.

Outcome	Groups	Subgroups	Number of Studies	Hazard/Risk Ratio [95%CI], *p*	Subgroups Heterogeneity
PFS	Geographic origin	Europe	7	0.418 [0.368, 0.474], *p* ≤ 0.001	*p* = 0.000
		America	4	0.677 [0.569, 0.805], *p* ≤ 0.001	
		Asia	2	0.643 [0.468, 0.859], *p =* 0.003	
	No. of patients	<100	7	0.417 [0.365, 0.476], *p* ≤ 0.001	*p* = 0.000
		≥100	6	0.635 [0.551, 0.732], *p* ≤ 0.001	
	Hypertension criteria	CTCAE V2.0	2	0.374 [0.295, 0.408], *p ≤* 0.001	*p* = 0.000
		CTCAE V3.0	6	0.662 [0.563, 0.778], *p* ≤ 0.001	
		CTCAE V4.0	3	0.659 [0.505, 0.860], *p =* 0.002	
		CTCAE V5.0	1	0.570 [0.403, 0.806], *p =* 0.001	
	Hypertension cut off in controls	Grade 0	6	0.724 [0.605, 0.867], *p* ≤ 0.001	*p* = 0.000
		Grade 1/2	6	0.432 [0.383, 0.488], *p* ≤ 0.001	
OS	Geographic origin	Europe	7	0.582 [0.476, 0.713], *p* ≤ 0.001	*p* = 0.001
		America	4	0.861 [0.714, 1.039], *p* = 0.119	
		Asia	1	0.349 [0.212, 0.575], *p* ≤ 0.001	
	No. of patients	<100	5	0.823 [0.662, 1.025], *p* = 0.081	*p* = 0.034
		≥100	7	0.612 [0.517, 0.723], *p* ≤ 0.001	
	Hypertension criteria	CTCAE V2.0	1	0.544 [0.347, 0.853], *p* = 0.002	*p =* 0.056
		CTCAE V3.0	7	0.769 [0.644, 0.918], *p* = 0.004	
		CTCAE V4.0	2	0.662 [0.487, 0.900], *p* = 0.008	
		CTCAE V5.0	1	0.320 [0.178, 0.574], *p* ≤ 0.001	
	Hypertension cut off in controls	Grade 0	6	0.746 [0.612, 0.910], *p* = 0.004	*p* = 0.477
		Grade 1/2	5	0.626 [0.512, 0.766], *p* ≤ 0.001	
ORR	Geographic origin	Europe	9	1.287 [1.108, 1.495], *p* = 0.001	NA
		America	0	NA	
		Asia	0	NA	
	No. of patients	<100	5	1.468 [1.197, 1.800], *p* ≤ 0.001	*p* = 0.063
		≥100	4	1.103 [0.885, 1.376], *p* = 0.383	
	Hypertension criteria	CTCAE V2.0	2	1.649 [1.110, 2.452], *p* = 0.013	*p* = 0.001
		CTCAE V3.0	6	1.155 [0.978, 1.364], *p* = 0.090	
		CTCAE V4.0	1	3.970 [1.981, 7.955], *p* ≤ 0.001	
	Hypertension cut off in controls	Grade 0	5	1.183 [0.970, 1.441], *p* = 0.097	*p* = 0.198
		Grade 1/2	4	1.443 [1.147, 1.815], *p =* 0.002	

**Table 4 jpm-14-00520-t004:** Leave-one-out sensitivity analysis of the included studies.

Outcome	Excluded Study	Hazard/Risk Ratio [95% CI]	*p*-Value
PFS	Budai et al. (2013) [17]	0.505 [0.456, 0.560]	<0.001
	Dionisio de Sousa et al. (2016) [18]	0.503 [0.456, 0.555]	<0.001
	De Stefano et al. (2011) [19]	0.506 [0.458, 0.558]	<0.001
	Dewdney et al. (2012) [20]	0.505 [0.458, 0.557]	<0.001
	Feliu et al. (2015) [21]	0.501 [0.454, 0.553]	<0.001
	Morita et al. (2013) [24]	0.500 [0.453, 0.551]	<0.001
	Osterlund et al. (2011) [25]	0.503 [0.455, 0.555]	<0.001
	Rattner et al. (2023) [27]	0.467 [0.421, 0.518]	<0.001
	Ryanne Wu et al. (2009) [28]	0.503 [0.456, 0.555]	<0.001
	Saito et al. (2023) [29]	0.502 [0.454, 0.555]	<0.001
	Scartozzi et al. (2009) [3]	0.630 [0.560, 0.708]	<0.001
	Sud et al. (2018) [30]	0.508 [0.460, 0.562]	<0.001
	Tahover et al. (2013) [31]	0.499 [0.451, 0.552]	<0.001
OSS	Budai et al. (2013) [17]	0.676 [0.583, 0.784]	<0.001
	Dionisio de Sousa et al. (2016) [18]	0.680 [0.591, 0.781]	<0.001
	De Stefano et al. (2011) [19]	0.680 [0.594, 0.779]	<0.001
	Dewdney et al. (2012) [20]	0.675 [0.590, 0.773]	<0.001
	Feliu et al. (2015) [21]	0.697 [0.607, 0.801]	<0.001
	Khoja et al. (2014) [22]	0.683 [0.596, 0.782]	<0.001
	Osterlund et al. (2011) [25]	0.709 [0.616, 0.816]	<0.001
	Rattner et al. (2023) [27]	0.652 [0.568, 0.0.748]	<0.001
	Ryanne Wu et al. (2009) [28]	0.609 [0.522, 0.710]	<0.001
	Saito et al. (2023) [29]	0.706 [0.615, 0.811]	<0.001
	Sud et al. (2018) [30]	0.678 [0.587, 0.782]	<0.001
	Tahover et al. (2013) [31]	0.671 [0.582, 0.774]	<0.001
ORR	Budai et al. (2013) [17]	1.434 [1.210, 1.700]	<0.001
	Dionisio de Sousa et al. (2016) [18]	1.218 [1.045, 1.421]	0.012
	De Stefano et al. (2011) [19]	1.339 [1.127, 1.591]	0.001
	Dewdney et al. (2012) [20]	1.334 [1.140, 1.561]	<0.001
	Feliu et al. (2015) [21]	1.282 [1.096, 1.500]	0.002
	Khoja et al. (2014) [22]	1.219 [1.040, 1.428]	0.014
	Mir et al. (2011) [23]	1.281 [1.094, 1.501]	0.002
	Osterlund et al. (2011) [25]	1.280 [1.098, 1.492]	0.002
	Scartozzi et al. (2009) [3]	1.245 [1.067, 1.452]	0.005

## Data Availability

The original contributions presented in the study are included in the article/Appendix A, further inquiries can be directed to the corresponding author.

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
