# Peer review of "Clinical Outcome of Colorectal Cancer Patients with Concomitant Hypertension: A Systematic Review and Meta-Analysis"

_jpm, 2024, doi:10.3390/jpm14050520_

Round 1
Reviewer 1 Report
Comments and Suggestions for Authors
1. Are any ethical considerations or patient safety issues addressed in this study, particularly regarding using patient data from the included articles?
2. How do these findings contribute to our understanding of the role of hypertension in colorectal cancer prognosis and treatment?
3. Discuss potential mechanisms by which hypertension might influence tumor biology and treatment response in metastatic colorectal cancer patients.
4. Elaborate on the novelty of your study and how it fills a potential research gap in metastatic colorectal cancer?
5. Did the authors observe any trends or patterns in treatment responses or survival outcomes among hypertensive patients that could provide insights into the prognostic significance of hypertension in this population?
6. Are there any notable trends or variations in hazard ratios among the included studies, and how do these findings contribute to our understanding of the impact of hypertension on clinical outcomes in metastatic colorectal cancer patients?
7. The conclusion mentions that elevated blood pressure is associated with better clinical outcomes, but are there any potential confounding factors or biases that could have influenced this association?
I recommend a major revision to further address these considerations and strengthen the manuscript.
Author Response
Reviewer 1
Dear Reviewer,
We want to thank the reviewer for the time spent analyzing our manuscript. We are sure that by addressing the points specified by the reviewer we can improve our current work.
- Are any ethical considerations or patient safety issues addressed in this study, particularly regarding using patient data from the included articles?
We thank the reviewer for pointing this out. Response: No, there is no ethical considerations or patients’ safety issues addressed in this study. Meta-analysis study did not require ethical statement because it used data already published by other studies that have got ethical clearance from participants.
- How do these findings contribute to our understanding of the role of hypertension in colorectal cancer prognosis and treatment?
Despite several studies explored the role of hypertension in colorectal cancer most of the studies concentrated only on Bevacizumab induced hypertension. We emerged in analyzing other anti-angiogenic agents as well. Our study had a more extensive approach providing an update in this clinical scenario.
- Discuss potential mechanisms by which hypertension might influence tumor biology and treatment response in metastatic colorectal cancer patients.
We thank the reviewer for the suggestion. Although insufficient data is available about the matter, we discussed the potential influence in lines 304-318.
- Elaborate on the novelty of your study and how it fills a potential research gap in metastatic colorectal cancer?
We thank the reviewer for the suggestion. We adjusted our manuscript to make a clear point about the novelty of the study in lines 325-334.
- Did the authors observe any trends or patterns in treatment responses or survival outcomes among hypertensive patients that could provide insights into the prognostic significance of hypertension in this population?
We thank the reviewer for the question. Response: Yes, we noticed trends in survival outcomes among hypertensive patients. Indeed, HTN could be a significant predictive factor of PFS, OS and ORR in patients with metastatic colorectal cancer.
- Are there any notable trends or variations in hazard ratios among the included studies, and how do these findings contribute to our understanding of the impact of hypertension on clinical outcomes in metastatic colorectal cancer patients?
Yes, there is a notable variation in hazard ratios among the included studies. The majority of studies revealed that HTN was associated with significant improvement in PFS, OS and ORR values, while some studies do not. However, the pooled analysis of all data suggested that HTN could be a significant predictive factor for these survival outcomes.
- The conclusion mentions that elevated blood pressure is associated with better clinical outcomes, but are there any potential confounding factors or biases that could have influenced this association?
We appreciate the reviewer’s input. Response: Yes of course, subgroup analysis showed that the geographic origin of studies, sample size, HTN criteria and HTN grade of controls were significant sources of heterogeneity that could have influenced this association.
I recommend a major revision to further address these considerations and strengthen the manuscript.
We thank the evaluator for the suggestion. The text has been revised by a team of professional medical editors and the text has been double-checked for any issues.
Reviewer 2 Report
Comments and Suggestions for Authors
The paper from Sur et al. investigates the role of blood pressure in patients with colorectal cancer and primary hypertension. The work is well written and globally sound, but I believe there are some reporting flaws that should be ironed out. Hereby my suggestions to the Authors:
1) L 91: the Authors mention several databases where the search was performed, but present only Pubmed Search string. Strategies for other databases should be reported aswell.
2) L 127: the Authors mentioned the inclusion of both prospective and retrospective studies, but used Newcastle-Ottawa Scale (NOS) for quality evaluation. Why? NOS is used for assessing nonrandomised studies, so the Authors should mention if this was used because all retrieved studies were as such.
3) L 160: "A fixed-effects design was used when I2 < 50% and p > 0.05; otherwise, a random-effects model was adopted [15]". This approach to I2 and modle choice is, unfortunately, a very common but very wrong misconception. The paper from Borenstein et al. the Authors seem to quote after this sentece does not support this at all: actually, it explicitely states that fixed-effects assumptions are high unlikely in most meta-analysis setting. Also, the choice of model to use (Fixed, Common or Random) should be made a priori based on the hypotesis to be tested, and not a posteriori after I2 calculation. Moreover, I2 is a measure of [statistical heterogeneity], but does not account for actual heterogeneity such as differences in study design\population\measures of outcome etc. As such, Random-effects models should be used by default in meta-analysis, and any usage of fixed models should be validly justified. Cfr. also https://training.cochrane.org/handbook/current/chapter-10, 10.1002/sim.1186, and 10.1002/sim.3478.
For the record, the studies actually included among the results came from 8 different countries, and this underlying difference in population violates the fixed effect assumption by its own.
4) As a random effect model is used anyway, prediction interval should also be reported, both narratively and in figures. Cfr. 10.1136/bmjopen-2015-010247.
5) L. 172: results from each database search (before pooling and duplicate removal) should be reported. This should be the case also for Fig. 1.
6) For all forest plots (Fig. 2-4): cfr. point 4).
7) For all forest plots (Fig. 2-4): I suggest not to use graphical log scale but simple untrasformed scale. there is no need of a scale that shows 0-100 if the highest Upper CI limit isn't even 8!
8) As asymmetry was identified , effect size estimation via trim-and-fill ahould also be performed. Cfr. 10.1111/j.0006-341x.2000.00455.x.
9) Subgroup analysis is well conducted, but it is clumsy to read by table; could you also put it into the main funnel plots figures (Fig 2-4)? its visualization would help the reader with correct interpretation.
Author Response
Reviewer 2
Dear Reviewer,
We want to thank the reviewer for the time spent evaluating our manuscript. We are sure that by answering the issues pointed out by the reviewer we can further improve our manuscript.
The paper from Sur et al. investigates the role of blood pressure in patients with colorectal cancer and primary hypertension. The work is well written and globally sound, but I believe there are some reporting flaws that should be ironed out. Hereby my suggestions to the Authors:
1) L 91: the Authors mention several databases where the search was performed, but present only Pubmed Search string. Strategies for other databases should be reported aswell.
Response: We added a supplementary file 1 that included search strategies for all databases.
2) L 127: the Authors mentioned the inclusion of both prospective and retrospective studies, but used Newcastle-Ottawa Scale (NOS) for quality evaluation. Why? NOS is used for assessing nonrandomised studies, so the Authors should mention if this was used because all retrieved studies were as such.
Response: Thank you for your remark. We mentioned that in the revised version.
3) L 160: "A fixed-effects design was used when I2 < 50% and p > 0.05; otherwise, a random-effects model was adopted [15]". This approach to I2 and modle choice is, unfortunately, a very common but very wrong misconception. The paper from Borenstein et al. the Authors seem to quote after this sentece does not support this at all: actually, it explicitely states that fixed-effects assumptions are high unlikely in most meta-analysis setting. Also, the choice of model to use (Fixed, Common or Random) should be made a priori based on the hypotesis to be tested, and not a posteriori after I2 calculation. Moreover, I2 is a measure of [statistical heterogeneity], but does not account for actual heterogeneity such as differences in study design\population\measures of outcome etc. As such, Random-effects models should be used by default in meta-analysis, and any usage of fixed models should be validly justified. Cfr. also https://training.cochrane.org/handbook/current/chapter-10, 10.1002/sim.1186, and 10.1002/sim.3478.
For the record, the studies actually included among the results came from 8 different countries, and this underlying difference in population violates the fixed effect assumption by its own.
Response: Thank you for your valuable input. We have corrected this issue in the revised manuscript.
4) As a random effect model is used anyway, prediction interval should also be reported, both narratively and in figures. Cfr. 10.1136/bmjopen-2015-010247.
Reponse: Unfortunately, we can’t report such information using Comprehensive meta-analysis software.
5) L. 172: results from each database search (before pooling and duplicate removal) should be reported. This should be the case also for Fig. 1.
Reponse; We thank the reviewer for the suggestion. We added this information in the revised version.
6) For all forest plots (Fig. 2-4): cfr. point 4).
Response: Unfortunately, we can’t report such information using Comprehensive meta-analysis software.
7) For all forest plots (Fig. 2-4): I suggest not to use graphical log scale but simple untrasformed scale. there is no need of a scale that shows 0-100 if the highest Upper CI limit isn't even 8!
Response; Thank you for the suggestions. We modified the manuscript accordingly.
8) As asymmetry was identified , effect size estimation via trim-and-fill ahould also be performed. Cfr. 10.1111/j.0006-341x.2000.00455.x.
Response: We modified accordingly in the revised version of our manuscript.
9) Subgroup analysis is well conducted, but it is clumsy to read by table; could you also put it into the main funnel plots figures (Fig 2-4)? its visualization would help the reader with correct interpretation.
Response: Subgroup analysis included 4 different groups (Geographic origin, No. of patients, Hypertension criteria, Hypertension cut off in controls), so we can’t include them all in the forest plots, we have to produce 4 forest plots for each outcome, which is not feasible.
Round 2
Reviewer 1 Report
Comments and Suggestions for Authors
The manuscript has been improved, and its acceptable for publication in the present form.
Reviewer 2 Report
Comments and Suggestions for Authors
The Authors answered to most of my previous comments. I have no further remarks.